

# Remote sensing techniques for automated marine mammals detection: a review of methods and current challenges

Esteban N. Rodofili[1], Vincent Lecours[1,2] and Michelle LaRue[3,4]

[1] School of Natural Resources and Environment, University of Florida, Gainesville, FL, United States of America
[2] School of Forest, Fisheries, and Geomatics Sciences, University of Florida, Gainesville, FL, United States of America
[3] School of Earth and Environment, University of Canterbury, Christchurch, New Zealand
[4] Department of Earth and Environmental Science, University of Minnesota, Minneapolis, MN, United States of America

Corresponding author
Esteban N. Rodofili, erodofili@ufl.edu

## ABSTRACT

Marine mammals are under pressure from multiple threats, such as global climate change, bycatch, and vessel collisions. In this context, more frequent and spatially extensive surveys for abundance and distribution studies are necessary to inform conservation efforts. Marine mammal surveys have been performed visually from land, ships, and aircraft. These methods can be costly, logistically challenging in remote locations, dangerous to researchers, and disturbing to the animals. The growing use of imagery from satellite and unoccupied aerial systems (UAS) can help address some of these challenges, complementing crewed surveys and allowing for more frequent and evenly distributed surveys, especially for remote locations. However, manual counts in satellite and UAS imagery remain time and labor intensive, but the automation of image analyses offers promising solutions. Here, we reviewed the literature for automated methods applied to detect marine mammals in satellite and UAS imagery. The performance of studies is quantitatively compared with metrics that evaluate false positives and false negatives from automated detection against manual counts of animals, which allows for a better assessment of the impact of miscounts in conservation contexts. In general, methods that relied solely on statistical differences in the spectral responses of animals and their surroundings performed worse than studies that used convolutional neural networks (CNN). Despite mixed results, CNN showed promise, and its use and evaluation should continue. Overall, while automation can reduce time and labor, more research is needed to improve the accuracy of automated counts. With the current state of knowledge, it is best to use semi-automated approaches that involve user revision of the output. These approaches currently enable the best tradeoff between time effort and detection accuracy. Based on our analysis, we identified thermal infrared UAS imagery as a future research avenue for marine mammal detection and also recommend the further exploration of object-based image analysis (OBIA). Our analysis also showed that past studies have focused on the automated detection of baleen whales and pinnipeds and that there is a gap in studies looking at toothed whales, polar bears, sirenians, and mustelids.

## INTRODUCTION

Marine mammals currently face various anthropogenic threats such as fishery bycatch, vessel strikes, competition for resources with commercial fisheries, noise pollution, bioaccumulation of pathogens and toxins, harmful algal blooms, and climate change (*Merrick, Silber & De Master, 2018*). In this context, surveying efforts to improve knowledge of marine mammal species' distributions and abundance are critical for their conservation. Surveys can be carried out from land, aircraft or ships for cetaceans (*Hiby & Hammond, 1989*). Pinnipeds have been usually counted from land-based viewing stations, although aerial photographs have also provided a viable alternative (*Moore, Forney & Weller, 2018*). Boat surveys around rookeries are also commonly performed (*e.g.*, *Arias-del Razo et al., 2016*; *Adame et al., 2017*). However, crewed aerial and vessel platforms can bias observations due to the animals' reactions to them (*Würsig et al., 1998*; *Born Riget, Dietz & Andriashek, 1999*; *Dawson et al., 2004*; *Luksenburg & Parsons, 2009*; *Hashim & Jaaman, 2011*). In addition, crewed aerial and boat surveys, as well as ground surveys, can be dangerous to researchers (*Sasse, 2003*; *Hodgson, Kelly & Peel, 2013*; *Gooday et al., 2018*), expensive, and logistically challenging for remote populations far away from an airstrip or a port (*Hodgson, Kelly & Peel, 2013*; *Gooday et al., 2018*; *Höschle et al., 2021*).

In this context, satellite and unoccupied aerial systems (UAS) imagery have surged as complementary tools to count individuals of various taxa of marine mammals (Sirenia, Ursidae, Pinnipedia, and Cetacea: cf. Table 1). These two platforms share the advantage of minimizing danger to researchers. Furthermore, the two platforms can provide still images researchers can revisit and use to cross-validate their counts with other researchers (*e.g.*, *Gonçalves, Spitzbart & Lynch, 2020*). In particular, very-high-resolution (VHR) satellite imagery (*i.e.*, sub-meter spatial resolution) allows monitoring marine mammals in remote areas (*LaRue, Stapleton & Anderson, 2017*) without any disturbance to wildlife (*LaRue et al., 2011*). As a result of a wide geographic coverage, VHR satellite images have the potential to fill knowledge gaps on distribution, abundance, density, and population trends of different marine mammal taxa (cf. Table 1) while supplementing data from field-based surveys and allowing for continued monitoring when in-person surveys are not possible (*Höschle et al., 2021*). A comparison between satellite and ship survey density estimates showed encouraging results for satellite imagery (*i.e., Bamford et al., 2020*). On the other hand, despite still requiring fieldwork and offering less area coverage than satellites, UAS allow researchers more freedom in terms of survey timing than satellites and permit coverage in regions of perennial overcast weather (*e.g., Goebel et al., 2015*). *Adame et al. (2017)* found their UAS survey to be more accurate than a traditional boat-based survey for the study of a California sea lion (*Zalophus californianus*) rookery. Furthermore, UAS collect multispectral imagery of higher resolution than satellites, at centimeter (*Johnston, 2019*) and even sub-centimeter spatial resolutions (*Raoult et al., 2020*), which is especially

relevant for smaller marine mammals like mustelids (*e.g.*, sea otters - *Enhydra lutris*). Moreover, UAS can provide thermal infrared (IR) imagery at sufficient resolution to detect marine mammals on land (*e.g., Seymour et al., 2017*), unlike satellite-based thermal IR, which offers too coarse of a resolution (*e.g., United States Geological Survey, 2020*).

Despite their advantages, satellite and UAS platforms are not panaceas. The cost of satellite imagery still constitutes a limiting factor for many studies (*Turner et al., 2015*; *Höschle et al., 2021*). UAS, on the other hand, can be disturbing to animals depending on their noise profile (*Pomeroy, O'Connor & Davies, 2015*). Furthermore, high-endurance UAS recommended to cover large regions (*Raoult et al., 2020*) are costly, and their use can be limited based on civil aviation regulations (*Fiori et al., 2017*). In such cases, satellites remain the most cost-effective solution to study large-bodied animals over large areas (*Johnston, 2019*). In addition to these issues that depend on costs, regulatory framework, and technological advancements of these platforms, manual counts of animals in remote sensing data are extremely labor and time-intensive for researchers over large areas (*Hollings et al., 2018*; *Höschle et al., 2021*). For example, the analysis of 425 km$^2$ (still a relatively small area in terms of marine mammal distribution) took 24 h to be analyzed with three replicates (eight hours per person) for the work of *Thums et al. (2018)*, corresponding to 1.13 min per km$^2$. Furthermore, *Cubaynes et al. (2019)* reported three hours and 20 min to scan 100 km$^2$ at a 1:1,500 m. scale (2 min per km$^2$), and *Charry et al. (2021)* reported an even slower rate of approximately 2.5 min per km$^2$ (at a 1:536 scale). While satellite images have allowed continental-scale abundance and distribution studies in remote locations (*e.g.*, *LaRue et al., 2019*; *LaRue et al., 2020*; *LaRue et al., 2021*), that work required crowdsourcing: *LaRue et al. (2020)* enlisted more than 325,000 volunteers to analyze 268,611 km$^2$ of images while a previous effort (*LaRue et al., 2019*) used more than 5,000 volunteers for the Ross Sea region in Antarctica. These broad-scale studies remain scarce for marine mammals: most work based on satellite images is performed over smaller areas such as individual islands (e.g., *LaRue et al., 2015*; *LaRue & Stapleton, 2018*).

Automation of image analysis reduces the level of effort and time needed, especially for large-scale projects (*Hollings et al., 2018*; *Charry et al., 2021*; *Höschle et al., 2021*). Even semi-automation in *Thums et al. (2018)* shortened 24 person-hours to 34 min to run the algorithm and an additional 20 min to revise sub-sections with whale detections and discard those that were false (a reduction of more than 96% of the analysis time). Automation could also help with errors caused by human fatigue from reviewing imagery (*Hodgson, Kelly & Peel, 2013*). As such, automation could unleash the full potential of VHR satellite images to cover more extensive study areas, which is ultimately a capability that distinguishes the use of satellite imagery from crewed surveys. Automating animal detection in UAS imagery has also been suggested to provide better survey data (*Hodgson, Peel & Kelly, 2017*). Just as for satellite imagery, automation would also reduce the labor and time necessary to count animals from UAS imagery (*Linchant et al., 2015*). Technologies for automated count of animals have emerged for areas difficult to access, in which ground counts are difficult, or for animals that are at low densities over large areas (*Hollings et al., 2018*). Marine mammals generally fit all these conditions, given their aquatic habitats and extended ranges, making them suitable candidates for the application of automated counts. However, automated

**Table 1  Marine mammal automated detection studies using satellite or UAS imagery.** Imagery type is based on testing imagery, although in several studies the training imagery was from the same source. Automated studies results were assessed with Eqs. (3), (4) and (5). Missed animals refer to Eq. (3), false animals refer to Eq. (4), and total deviation refers to Eq. (5). For automation results assessment calculations, see Eqs. (3), (4) and (5), and Table S1.

| Study | Taxa | Platform; altitude (if UAS); imagery type (spatial resolution) | Automated method | Automation results assessment |
|---|---|---|---|---|
| *Mejias Alvarez et al. (2013)* | Dugongs (*Dugong dugon*) (Sirenia) | UAS (ScanEagle) with Nikon 12 MP. digital SLR camera and 50 mm. lens and polarizing filter; 500/750/1000 ft.; RGB (ground sampling resolution not found) | Morphological based detection, segmentation, shape profiling on saturation channel | Not enough data was found to calculate Eqs. (3), (4) and (5) |
| *Maire et al. (2013)* | Dugongs (*Dugong dugon*) (Sirenia) | UAS (not specified) with Nikon 12 MP. digital SLR camera and 50 mm. lens and polarizing filter; 500/750/1000 ft.; RGB (ground sampling resolution not found) | Color and morphological filters, segmentation and shape analysis | Not enough data was found to calculate Eqs. (3), (4) and (5) |
| *Fretwell, Staniland & Forcada (2014)* | Southern right whales (*Eubalena australis*) (Mysticeti) | Satellite (WorldView-2); panchromatic (0.5 m.) and multispectral (~2 m. but pan-sharpened) | Unsupervised classification, supervised classification and histogram thresholding | Supervised classification: no meaningful results Unsupervised classification Kmeans: 41.76% (missed), 53.85% (false), 95.6% (total deviation) Histogram thresholding Band 5: 15.38% (missed), 26.37% (false), 41.76% (total deviation) (Best variants within methods chosen based on Eq. (5) for total signals, not by probable, possible and band 5 manual detections) |
| *LaRue et al. (2015)* | Polar bears (*Ursus maritimus*) (Ursidae) | Satellite (WorldView-2/QuickBird)**; panchromatic (0.5-0.65 m.)* | Supervised classification and image differencing | Supervised classification: unsuccessful. Image differencing: not enough data was found to calculate Eqs. (3), (4) and (5). |
| *Seymour et al. (2017)* | Grey seals (*Halichoerus grypus*) (Pinnipedia) | UAS (senseFly eBee) with 12 MP. RGB Canon S110 camera and 640 × 512-pixel thermal IR senseFly LLC, Thermomapper camera; altitude not found; RGB (3 cm.) and thermal IR (8 cm.) | ArcGis Model based on temperature, size and shape | Pups: Saddle Island (simple): 6.45% (missed), 13.55% (false), 20% (total deviation) Saddle Island (Complex): 7.04% (missed), 13.9% (false), 20.94% (total deviation) Adults: Saddle Island (simple): 1% (missed), 23.38% (false), 24.38% (total deviation) Saddle Island (Complex): 0.98% (missed), 49.02% (false), 50% (total deviation) (only models from prediction site) |

| Study | Taxa | Platform; altitude (if UAS); imagery type (spatial resolution) | Automated method | Automation results assessment |
|---|---|---|---|---|
| *Thums et al. (2018)* | Humpback whales (*Megaptera novaeangliae*) (Mysticeti) | Satellite (WorldView-2/WorldView-3); panchromatic (0.4 m.) and multispectral (but pansharpened) / panchromatic (0.31 m.) | Unsupervised classification and supervised classification (in Worldview 2 imagery) and semi-automated algorithm based on shape (in Worldview-3 imagery) | Unsupervised classification and supervised classification: not enough data found for Eqs. (3), (4) and (5). Shape algorithm: 0% (missed), 205.8% (false), 205.8% (total deviation) (calves and mothers together, all images combined) |
| *Gray et al. (2019)* | Blue whales (*Balaenoptera musculus*) and humpback whales (*Megaptera novaeangliae*) and Antarctic minke whales (*Balaenoptera bonaerensis*) (Mysticeti) | UAS (FreeFly Alta 6/LemHex-44) with Sony a5100 camera with 50 mm. Focal length lens, 23.5 × 15.6 mm. Sensor size and 6000 × 4000 pixel resolution; 30–80 m.; RGB (see equation in study for ground sampling distance) | CNN (deep learning) | 0% (missed), 1.72% (false), 1.72% (total deviation) (only for whale recognition, not species) |
| *Borowicz et al. (2019)* | Southern right whales (*Eubalaena australis*) and humpback whales (*Megaptera novaeangliae*) (Mysticeti) | Satellite (WorldView-3); multispectral (1.24 m. but pansharpened to 0.31 m.) | CNN (deep learning) | 0% (missed), 271.88% (false), 271.88% (total deviation) (best model chosen by authors) |
| *Guirado et al. (2019)* | Various | Google Earth Imagery (included USGS aerial/WorldView-3/QuickBird-2/GeoEye-1/SPOT-6/WorldView-2); RGB (0.15 m./0.31 m. panchromatic–1.24 m. multispectral/0.61 m. panchromatic–2.5 m. multispectral/0.46 m. panchromatic–1.84 m. multispectral/1.5 m. panchromatic–6 m. multispectral/0.46 m. panchromatic–1.84 m. multispectral) | CNN (deep learning) | Detection CNN: 20.59% (missed), 7.35% (false), 27.94% (total deviation). Count CNN: 11.43% (missed), 4.29% (false), 15.71% (total deviation) (all locations totals) |

**Table 1** (*continued*)

| Study | Taxa | Platform; altitude (if UAS); imagery type (spatial resolution) | Automated method | Automation results assessment |
|---|---|---|---|---|
| *Cubaynes (2019)* | Southern right whales (*Eubalaena australis*) (Mysticeti) | Satellite (GeoEye-1); multispectral (1.65 m. but pansharpened to 0.41 m.) | Unsupervised classification, supervised classification, thresholding and OBIA | Unsupervised classification Isodata: 56.82% (missed), 140.91% (false), 197.73% (total deviation). Supervised classification (maximum likelihood): 9.09% (missed), 3.41% (false), 12.5% (total deviation). Thresholding (only NIR1): 34.09% (missed), 194.32% (false), 228.41% (total deviation). OBIA: 27.27% (missed), 453.41% (false), 480.68% (total deviation). (for total whales, not by definite, probable and possible) |
| *Gonçalves, Spitzbart & Lynch (2020)* | Crabeater seals (*Lobodon carcinophaga*), Weddell seals (*Leptonychotes weddellii*), leopard seals (*Hydrurga leptonyx*) and Ross seals (*Omnatophoca rossii*) (Pinnipedia) | Satellite (WorldView-3); panchromatic (0.3 m.) | CNN (deep learning) | SealNet: 69.78% (missed), 51.71% (false), 121.49% (total deviation) (total for all scenes) (only model with best F1 for testing) |
| *Zinglersen et al. (2020)* | Walruses (*Odobenus rosmarus*) (Pinnipedia) | Satellite (Pléiades 1 A/B/WorldView-2/WorldView-3); multispectral (2 m. but pansharpened to 0.5 m./1.84 m. but pansharpened to 0.46 m./1.24 m. but pansharpened to 0.31 m.) | OBIA | Not enough data found for Eqs. (3), (4) and (5). |
| *Dujon et al. (2021)* | Australian fur seal (*Arctocephalus pusillus*) (Pinnipedia) | UAS (DJI Phantom 4 Professional™ V2) with built-in camera; 35 m.; RGB (ground sampling resolution not found) | CNN (Deep learning) | Not enough data found for Eqs. (3), (4) and (5). |

**Notes.**
*Information obtained from *Hollings et al. (2018)*
**Information obtained from *Cubaynes (2019)*

detection in UAS imagery has been more often applied to terrestrial mammals and birds than to marine mammals (*Corcoran et al., 2021*), and *Hollings et al. (2018)* report a similar pattern for automated analysis of remote sensing imagery in general.

Automated analysis of remote sensing imagery can be divided into pixel or object-based methods, depending on the unit of analysis (*Blaschke et al., 2014*; *Wang, Shao & Yue, 2019*). While pixel-based methods (*e.g.*, unsupervised or supervised classification and thresholding) classify individual pixels into animal or background classes based on spectral information or texture, object-based methods classify groups of homogeneous pixels

into the different classes, which allows for the use of other variables, such as size and shape (*Blaschke et al., 2014*; *Wang, Shao & Yue, 2019*). This contribution reviews the use of different methods of automated analysis of VHR satellite and UAS imagery for marine mammal detection, which is a current gap as previously-published reviews are dedicated to wildlife in general, either to their automated detection in UAS (*i.e.*, *Corcoran et al., 2021*) or manual and automatic detection in UAS, satellite and aerial imagery (*i.e.*, *Hollings et al., 2018*; *Wang, Shao & Yue, 2019*). The detection of marine mammals in their natural habitats presents different challenges that warrant their own review. The analysis we present also adds to contributions that focused on using satellite (*LaRue, Stapleton & Anderson, 2017*; *Höschle et al., 2021*) and UAS (*Koski, Abgrall & Yazvenko, 2010*; *Smith et al., 2016*; *Fiori et al., 2017*; *Johnston, 2019*; *Schofield et al., 2019*; *Raoult et al., 2020*) imagery to study marine wildlife separately. This review aims to inform the multidisciplinary community of remote sensing experts, biologists, ecologists, and managers working on marine mammal research and conservation as to which automated methods and imagery type show more promise, but also which taxa have been overlooked, practical alternatives to ensure time-saving and accurate counts, and steps necessary to progress from individual studies to the regional-scale initiatives that conservation needs. Our analysis is also summarized in Table 1, which improves upon the work of *Hollings et al. (2018)* and *Cubaynes (2019)* by the application of uniform metrics for evaluating detection accuracy that allow comparing the results of existing studies directly and for a more straightforward assessment of over- and undercounts in a conservation context.

## SURVEY METHODOLOGY

The search engine of the University of Florida Libraries (https://uflib.ufl.edu/find/) Primo database was used repeatedly between August 2019 to January 2022 to identify peer-reviewed literature, conference proceedings, grey literature, and theses and dissertations about combinations of keywords such as "marine mammals", "remote sensing", "UAV", "UAS", "satellite" and "images". The Primo database searches across the majority of the authoritative electronic and print resources subscribed to and purchased by the University of Florida Libraries (*UF George A. Smathers Libraries, 2022*). The literature cited in these studies was then explored further if relevant to image analysis for marine mammal detection, either through Primo or Google Scholar. We excluded studies of detection of marine mammal groups (*e.g.*, *Burn & Cody, 2005*; *Fischbach & Douglas, 2021*), traces and holes (*e.g.*, *Platonov, Mordvintsev & Rozhnov, 2013*), carcasses (*e.g.*, *Fretwell et al., 2019*; *Clarke et al., 2021*), and works that compared automated detection in separate datasets of different spatial resolutions (*e.g.*, *Fischbach & Douglas, 2021*; *Corrêa et al., 2022*).

## CURRENT TRENDS IN MARINE MAMMAL AUTOMATED DETECTION

We found 13 studies on automated detection of marine mammals ranging from 2013 to 2021, of which five used UAS imagery and eight used satellite imagery (Table 1). While many different automation approaches have been tested, most studies used pixel-based

approaches (as opposed to object-based workflows). The early attempts in pixel-based automation used spectral information through unsupervised classifications, supervised classifications, histogram thresholding, and image differencing (Table 1), for example for the classification of Southern right whales (*Eubalaena australis*) (*Fretwell, Staniland & Forcada, 2014*) and polar bears (*Ursus maritimus*) (*LaRue et al., 2015*). Histogram thresholding consists of maximizing the class of interest's signal to reduce the amount of noise (*Fretwell, Staniland & Forcada, 2014*). Automated image differencing is a process to detect differences in pixels between satellite images collected at different times (*Singh, 1989*; *Lu et al., 2004*; *LaRue et al., 2015*). These methods proved not to be very effective in providing accurate counts (cf. Table 1). This aligns with recommendations from *Cubaynes et al. (2019)* to adopt approaches that can use more information (*e.g.*, topological, geometrical, or textural information) than only spectral responses, such as deep learning or object-based image analysis (OBIA).

Convolutional neural networks (CNN), a deep learning approach, have been increasingly used to analyze satellite and UAS images of whale hotspots (*Borowicz et al., 2019*; *Guirado et al., 2019*; *Gray et al., 2019*) and pinnipeds (*Gonçalves, Spitzbart & Lynch, 2020*; *Dujon et al., 2021*) (Table 1). CNN has had mixed results so far, with relative success when applied to satellite (*Guirado et al., 2019*) and UAS (*Gray et al., 2019*) images of whales. Of note, CNN proved able to distinguish between different species in *Gray et al. (2019)*, something which had not yet been achieved from satellite images (*Höschle et al., 2021*). However, not all CNN applications for whale detection were successful (*Borowicz et al., 2019*), and studies that applied CNN for the detection of pinnipeds in satellite (*Gonçalves, Spitzbart & Lynch, 2020*) and UAS (*Dujon et al., 2021*) imagery did not show high accuracies. *Gonçalves, Spitzbart & Lynch (2020)* reported the highest precision and recall among different scenes for their SealNet model as 0.519 and 0.377, respectively, in their crabeater seals (*Lobodon carcinophaga*), Weddell seals (*Leptonychotes weddellii*), leopard seals (*Hydrurga leptonyx*) and Ross seals (*Omnatophoca rossii*) study. *Dujon et al. (2021)* reported a 0.27 precision for Australian fur seal (*Arctocephalus pusillus*) detection. *Borowicz et al. (2019)* suggested including confounding classes such as ships, rocks, and land to minimize false positives and improve CNN accuracy. This worked relatively well for *Guirado et al. (2019)* but did not provide more accurate counts for *Gonçalves, Spitzbart & Lynch (2020)*. Another way to improve detection accuracy was the application of a two-step CNN that first detects whale presence in individual tiles of an image collection and then locates and counts whales in tiles that had them (*Guirado et al., 2019*). A valuable insight the authors offer about their two-step approach is that they find better results than with one detection model alone, as a result of the first CNN filtering false positives and therefore allowing the second CNN to count whales more accurately.

CNN studies have also sparked a discussion on how marine mammal behavior impacts detection abilities. For example, *Dujon et al. (2021)* developed a CNN capable of detecting, as mentioned before, Australian fur seals (*Arctocephalus pusillus*), but also loggerhead sea turtles (*Caretta caretta*), and Australian gannets (*Morus serrator*) in UAS imagery from separate sites. The authors reference the aggregation of pups in creches as one of the factors that made seal detection difficult compared to that of the other taxa analyzed, up to the

point that the CNN was not able to distinguish between multiple aggregated pups. Whales can also offer particular detection challenges based on their behavior. For instance, *Guirado et al. (2019)* concluded that more than 90% of true positives of their detection CNN were whales in blowing, breaching, peduncle emerged, or logging behavior, whereas 33% of false negatives were submerged whales—which happened to be the most frequent behavior observed. The remaining 66% of false negatives were spy-hopping whales (*Guirado et al., 2019*). *Cubaynes et al. (2019)* suggested that some species with less acrobatic behavior were easier to detect. This is not that clear in *Guirado et al.'s (2019)* findings, as some whale behaviors which could be characterized as acrobatic, *i.e.*, with their peduncle emerged and breaching, were part of most of the true positive cases along with blowing and logging. However, they recognized that the spyhopping behavior compromised the CNN's ability to detect whales.

In terms of detectability and accuracy evaluation, while *Guirado et al. (2019)* and *Cubaynes et al. (2019)* discussed the influence of behaviors displayed on the surface or submerged—but still visible—on the detectability of animals, there remains the issue of animals submerged to the point of not being visible. In general, studies that detect marine mammals automatically in imagery evaluate the accuracy of their automation methods by comparing the animals captured by the computer with the ones counted manually. This approach is based on obtaining a count of the detectable animals, *i.e.*, only those visible in the imagery. This is a common limitation of visual surveys from land, ships, aircraft, or using UAS or satellite imagery, and especially in marine mammal taxa that are permanently in the water (*i.e.*, sirenians and cetaceans). While, as mentioned before, UAS and satellite imagery have the advantage of allowing surveyors to go over them many times and cross-validate their counts with other researchers—helping to minimize perception bias compared to field surveys from land, ships or aircraft—they are still visual methods not able to detect animals submerged to the point of not being visible. Consequently, it is worth noting that while this review is focused on the automated detection of marine mammals in imagery to obtain animal counts as a first step, further steps are needed after the counts to calculate abundance or density estimates. For example, there is a requirement to make adjustements for submerged animals not showing in the imagery—animals available in the study area but not at the surface (not detectable), as *Bamford et al. (2020)* did for whale densities calculated from satellite imagery. These adjustments could involve additional information obtained through tracking devices, such as surfacing and submersion times (*e.g.*, *Bamford et al., 2020*).

The general approach presented above—comparing objects detected automatically with user observations assumed to correspond to all animals detectable in the image—aligns with evaluation methods commonly used in remote sensing. When adopting this approach, false negatives correspond to detectable animals missed by the computer. In contrast, false positives refer to anything that is not the detection target, such as water or rocks, and is classified wrongly as a marine mammal by the computer. False negatives thus contribute to an undercount of marine mammals, and false positives contribute to an overcount. However, there is currently a lack of consistency in accuracy metrics to evaluate automated methods across animal detection studies (*Hollings et al., 2018*). Furthermore,

when comparing automated detections with manual counts to evaluate detection accuracy, some studies defined the F1 index metric differently. For example, *Borowicz et al. (2019)* used Eq. (1) and *Gonçalves, Spitzbart & Lynch (2020)* used Eq. (2). This can be highly problematic when comparing different approaches.

$$F1 = \frac{2 * \text{precision} * \text{recall}}{\text{precision} + \text{recall}}. \tag{1}$$

Equation 1: F1 index as defined by *Borowicz et al. (2019)*.

$$F1 = \text{precision} * \text{recall}. \tag{2}$$

Equation 2: F1 index as defined by *Gonçalves, Spitzbart & Lynch (2020)*.

# FUTURE DIRECTIONS IN MARINE MAMMAL AUTOMATED DETECTION

## Object-based image analysis (OBIA)

OBIA has been less explored for marine mammal detection compared to pixel-based methods. While both pixel- and object-based approaches may consider more variables than spectral information (*e.g.*, texture) (*Cubaynes et al., 2019*), they are fundamentally different in their units of analysis (*Blaschke et al., 2014*). OBIA first segments the imagery, dividing it into spatially continuous objects whose internal heterogeneity is less than the heterogeneity of their neighbors (*Blaschke et al., 2014*). This produces scale-dependent, potentially more meaningful objects, or segments, made of relatively homogeneous pixels. The classification algorithm selected is then applied to the objects rather than the individual pixels making them up. In addition, OBIA provides more information than only the spectral response to inform the classification: for instance, information about the objects themselves, such as size, shape, relative or absolute location, boundary conditions, and topological relationships, can be integrated as features within the classification, something that cannot be done in pixel-based classifications (*Blaschke et al., 2014*). In theory, it would make sense that marine mammals be detected as homogeneous objects in remote sensing imagery.

Some studies have used object features, such as shape (*i.e.*, *Seymour et al., 2017*; *Thums et al., 2018*), and a study used highlighting of elliptical features, segmentation, and feature extraction, such as length and area, from blobs (*i.e.*, *Mejias Alvarez et al., 2013*). Another work by *Maire et al. (2013)* used shape from blobs obtained after segmentation with a prior determination of regions of interest using color and morphological filters. Results have not been conclusive: while the work of *Seymour et al. (2017)* proved successful in detecting pinnipeds in thermal IR imagery, *Thums et al. (2018)*, *Mejias Alvarez et al. (2013)* and *Maire et al. (2013)* did not get as good results in detecting a baleen whale species and a Sirenia species, respectively (cf. Table 1). Furthermore, OBIA was tested with mixed results. For Southern right whale (*Eubalaena australis*), *Cubaynes (2019)* had a moderate amount of missed (27.27%) animals, but a considerably high number of false positives (453.41%) (Table 1). *Zinglersen et al. (2020)* obtained a good object definition for walruses (*Odobenus rosmarus*), but this was dependent on site heterogeneity (*i.e.*, the relative composition of sand and snow impacted the analysis).

OBIA has not been used extensively to detect other animals either, leaving pixel-based studies as predominant. While OBIA has been suggested to perform better than pixel-based analyses in land classification, that is still not clear in the examples of OBIA for counting animals (see *Hollings et al., 2018*). It is also worth noting that OBIA can be computationally time-intensive (*Hollings et al., 2018*). Furthermore, there is a lack of an obvious method to measure segmentation accuracy and sometimes real-life objects in the image do not match the segments obtained, making it difficult to assess classification accuracy (*Ye, Pontius & Rakshit, 2018*).

## Thermal infrared (IR) imagery

Another resource worth exploring in the future for marine mammal automated image analysis is the use of thermal IR. The combined use of shape, size, and temperature captured from thermal IR allowed *Seymour et al. (2017)* to distinguish pinniped pups from adults and count animals in aggregations, which was a recognized challenge for this taxon. *Hollings et al. (2018)* argued that a high contrast between animals and their surroundings is particularly important for automation methods, which is consistent with *Cubaynes et al. (2019)* findings. Based on these arguments, thermal IR UAS imagery is promising for marine mammal automated detection as it should assist with distinguishing marine mammals from confounding objects or like-colored surfaces. While *Hollings et al. (2018)* found that automation attempts to count animals have shown reasonably high accuracy in areas that are small relative to a species' range and/or homogenous environments such as ice, the accuracy assessment metrics in Table 1 for studies that match this description suggest this may not be necessarily the case for marine mammal detection (see *Fretwell, Staniland & Forcada, 2014*; *Borowicz et al., 2019*; *Gonçalves, Spitzbart & Lynch, 2020* in Table 1). For these cases, thermal IR may be a tool worth exploring to increase the contrast between the animals and the background. However, thermal IR has only been used for automated detection of pinnipeds over land (*i.e.*, *Seymour et al., 2017*), and as such this recommendation can only be supported by research results in such settings. The potential of automated analyses on UAS thermal IR data for taxa that are in the water at all times (*i.e.*, cetaceans and sireneans) remains a knowledge gap at this time.

## Evaluation metrics

Here we assess the number of false negatives in the different studies of Table 1 using the miss rate, or false negative rate (FNR; Eq. (3)). We consider that this metric provides a straightforward assessment of false negatives—the undercount—in the context of conservation and management, as its denominator is the total number of animals that could have been detected (using the manual count as a reference). In order to have a metric that assesses false positives—the overcount—over the total number of animals that could have been detected too, we used a false positive over detectable rate (FPDR; Eq. (4)). In the context of a remote sensing application to inform conservation efforts, we considered it may be helpful to measure the undercount (all detectable animals missed by the computer) or the overcount (all false computer detections) against the actual animals detectable in the image, for a faster and direct assessment of how reliable an automated method would

be if adopted in conservation planning, and what the impact of its errors would be over counts later used in abundance estimations.

$$(1 - \text{recall}) \times 100 = \frac{FN}{TP + FN} \times 100 = \frac{FN}{\text{Detectable animals}} \times 100. \qquad (3)$$

False negative rate (Equation 3): ratio between the number of animals missed by an automated method (numerator) and the number of animals detectable in the image (the animals counted manually) (denominator), presented as a percentage. FN: false negatives. TP: true positives.

$$\frac{FP}{TP + FN} \times 100 = \frac{FP}{\text{Detectable animals}} \times 100. \qquad (4)$$

False positive over detectable rate (Eq. (4)): ratio between the number of objects wrongly classified as animals by an automated method (numerator) and the number of animals detectable in the image (the animals counted manually) (denominator), presented as a percentage. FP: false positives. FN: false negatives. TP: true positives.

To summarize these two metrics into one, in the same way the F1 index integrates precision and recall, we suggest the automated count deviation (Eq. (5)). The automated count deviation (Eq. (5)) adds false negatives to false positives and measures them against the detectable animals too, to help in conservation and management, as neither false positives nor negatives are desirable in that context. The FNR (Eq. (3)), FPDR (Eq. (4)) and automated count deviation (Eq. (5)) can also be used in combination with recall, precision and the F1 index for a more comprehensive assessment of performance.

Furthermore, we note that even though manual counts are taken as a reference of the animals detectable in the image, this is not an infallible strategy. The effects of environmental conditions, UAS flight-related variables, and the angle of imagery capture on manual detection certainty of different species should continue to be investigated (see *Aniceto et al., 2018*; *Subhan et al., 2019*). Furthermore, the subclassification of manual detections into probable or possible (*e.g.*, *Fretwell, Staniland & Forcada, 2014*) together with cross-validated manual counts among researchers (*e.g.*, *Gonçalves, Spitzbart & Lynch, 2020*) can help in attaining the best manual detection reference possible.

$$\frac{FN + FP}{TP + FN} \times 100 = \frac{FN + FP}{\text{Detectable animals}} \times 100. \qquad (5)$$

Automated count deviation (Eq. (5)): ratio between the sum of animals missed and the objects wrongly classified as animals by an automated method (numerator) and the animals detectable in the image (the animals counted manually) (denominator), presented as a percentage. FN: false negatives. FP: false positives. TP: true positives.

## DISCUSSION

The current threats marine mammals face justify the need for their populations to be monitored through spatially extensive and frequent surveys for abundance and distribution studies. Satellite and UAS imagery can help address these challenges and complement crewed surveys. Satellite images can provide extensive coverage, enabling even continental-scale counts to use for abundance and distribution estimations in remote locations (*e.g.*,

LaRue et al., 2019; LaRue et al., 2020; LaRue et al., 2021), making sampling efforts more globally even. Although UAS provide less spatial coverage than satellite images, they can allow more control over survey timing and surveying in overcast conditions, as well as higher spatial resolution, which can be of use for smaller marine mammals like sea otters (*Enhydra lutris*) or dolphins of the *Cephalorhynchus* genus. Nevertheless, the manual analysis of satellite and UAS imagery is time- and effort-intensive, for which automation can help (*Linchant et al., 2015*; *Hollings et al., 2018*; *Charry et al., 2021*; *Höschle et al., 2021*). While *Thums et al. (2018)* provided a comparative measure of time saved by their workflow, we recommend future automated marine mammal studies do the same, *i.e.*, that they evaluate how efficient different automated methods are in saving time and effort compared to a manual approach.

We also acknowledge the current challenges of automated approaches in matching the animals present in the imagery. In the studies we analyzed, automation efforts thus far have shown considerable deviation of their detections over the animals manually counted, in particular when it came to false positives (cf. Table 1). This is especially concerning in a conservation assessment context, as such an error could lead to the lifting of protection measures when they should stay in place, and therefore calls for future research to improve detection methods. Until automated methods are better, studies on marine mammal detection should compromise with semi-automated approaches that involve a user review (*Höschle et al., 2021*). However, one positive aspect is that even if a manual review is needed, a semi-automated method that yields image sub-sections or thumbnails that need to be revised manually afterwards can still shorten the overall revision time, as we have seen from *Thums et al. (2018)*. *Dujon et al. (2021)* also recommend semi-automation in the sense of leaving full automation for situations of animals that show optimal conditions for such analysis, and doing manual detections in more challenging settings. Furthermore, the authors recognized that while their CNN could be used in different species even within a single site, a sole algorithm would not be able to capture all components of different species optimally as a result of morphology, spacing behavior, and habitat (*Dujon et al., 2021*).

Here, we have reviewed the automated approaches for detecting marine mammals through satellite and UAS images. We observed that the automated methods reliant only on spectral information did not show high-performance (*e.g., Fretwell, Staniland & Forcada, 2014*; *LaRue et al., 2015*), and spectral contrast between baleen whales and their surroundings were found to be insufficient for automation, highlighting the need for other methods that can accommodate more information, such as deep learning and OBIA (*Cubaynes et al., 2019*). Deep learning showed high performance in some studies (*i.e., Guirado et al., 2019*; *Gray et al., 2019*) and even allowed for distinguishing between whale species (*i.e., Gray et al., 2019*), although this has not yet been achieved in satellite images (*Höschle et al., 2021*). Nevertheless, not all deep learning attempts were successful, and some attempts required hundreds (*e.g., Gray et al., 2019*) or thousands (*e.g., Guirado et al., 2019*) of images for training. An extra complication is that labeled satellite images are currently limited, highlighting the need for shared training datasets (*Höschle et al., 2021*). Overall, CNN shows promise, and future research should continue to improve on the studies already conducted. For instance, CNNs only successful attempts thus far have

been in whales—although the contrary is not true—for which further studies are due to see if the method's results in pinnipeds can be improved, and to evaluate the method in other smaller marine mammal taxa. The number of studies using CNN is still low, and more studies need to be undertaken to identify which components of workflows improve accuracy. For example, the use of more confounding classes seemed useful for *Guirado et al. (2019)* but was not a guarantee of success for *Gonçalves, Spitzbart & Lynch (2020)*.

We agree with *Hollings et al. (2018)* in that less costly images and software are necessary to drive the field forward. The use of free platforms for image analysis, such as Google Earth Engine, could provide a solution by also providing greater computational power to researchers. We recommend the development of semi-automated methods that can be applied to the simultaneous detection of multiple species sharing a common habitat. Therefore, free, regional, or even continental initiatives that combine training data for recognition of different species, through the cooperation among different research groups, government agencies or nonprofit organizations, are what is now necessary to provide a useful tool for conservation managers to obtain counts for abundances and distribution studies. Moreover, with readily available and accessible satellite image coverage, *Abileah (2001)* and *Cubaynes et al. (2019)* recommendations to use satellite imagery to study migrations could be finally implemented at a large scale, and object-based methods could provide exclusive applications, such as the detection of swimming directions of different individuals. This could complement the use of tagging devices and lead to mapping migration routes for policy measures aiming to reduce collisions and bycatch. Furthermore, accessible satellite image coverage combined with the speed of automation could help improve distribution estimations, discover unknown feeding and calving grounds, and keep track of distribution shifts due to climate change. Moreover, it is worth considering baleen whales have been the focus of most automated studies, and although some automated studies were applied to pinnipeds, sirenians, and polar bears, more studies are needed to evaluate different platforms, automation methods, or spectral bands for these other taxa. Automated studies on toothed whales and mustelids are also due. Here we focused on automated detection of living marine mammal individuals in VHR satellite and UAS imagery, but it is noteworthy that other promising satellite and UAS image analysis applications (automated and manual) exist and focus for example on detecting marine mammal groups (*e.g.*, *Burn & Cody, 2005*; *Fischbach & Douglas, 2021*), traces and holes (*e.g.*, *Platonov, Mordvintsev & Rozhnov, 2013*), carcasses (*e.g.*, *Fretwell et al., 2019*; *Clarke et al., 2021*), and on comparing detection success from imagery of different spatial resolutions (*e.g.*, *Fischbach & Douglas, 2021*; *Corrêa et al., 2022*).

Regarding new promising directions to explore, OBIA has been recommended for automated analysis (*Cubaynes et al., 2019*), but thus far pixel-based methods have been more thoroughly explored and the OBIA studies or the studies using object features have shown mixed results (*e.g.*, *Mejias Alvarez et al., 2013*; *Maire et al., 2013*; *Seymour et al., 2017*; *Thums et al., 2018*; *Cubaynes, 2019*; *Zinglersen et al., 2020*). Moreover, UAS offer data at sufficient resolution in the thermal IR, which has proven useful in distinguishing between pinniped pups and adults and counting individuals in aggregations (*i.e.*, *Seymour et al., 2017*). The use of multispectral imagery has been prevalent among studies thus far,

and we recommend further exploring the use of thermal IR as a solution to distinguish animals from similarly-colored surroundings, an issue spanning different taxa and surfaces (*e.g., LaRue et al., 2015*; *Cubaynes et al., 2019*; *LaRue et al., 2020*), or even to study marine mammals nighttime behavior, or during the polar night (*Boehme et al., 2016*). Furthermore, we decided to use the FNR (Eq. (3)), the FPDR (Eq. (4)) and the automated count deviation (Eq. (5)) proposed as a more straightforward means of assessing how reliable an automated method would be if adopted in conservation planning, and what the impact of its errors would be over counts later used in abundance estimations.

Marine mammal abundance and distribution studies can be well served from a combination of different surveying methods. For instance, UAS and satellite images, airplane and vessel surveys, to detect animals in the surface; passive acoustic monitoring, to detect submerged animals' vocalizations (*e.g., Davis et al., 2017*; *Stanistreet et al., 2018*); and satellite tracking, to obtain detailed movement patterns (*e.g., Horton et al., 2017*; *Horton et al., 2020*). In the last decade, studies using satellite and UAS imagery for marine mammal surveys have grown considerably in number, but lag behind the number of automated studies on other taxa. This may occur because of the logistical complexity of conducting UAS surveys at sea as opposed to on land (*e.g.,* launching and landing sites, weather conditions), as well as the relative scarcity of archival satellite imagery over sea compared to terrestrial habitats, leaving researchers only with more expensive tasking options. Automated methods have the potential to complement airplane and vessel marine mammal surveys, helping overcome old limitations, such as higher coverage in remote regions. However, to do that, satellite imagery needs to become more available and affordable further away from the coast, and high-endurance UAS need to become more affordable too, while their regulation needs to be flexibilized. In this context, joint initiatives for further research are necessary to not only to exchange imagery, but also to develop automated methods that both save researchers time and provide accurate counts, while being versatile enough to work in different settings and distinguish coexistent species.

## ACKNOWLEDGEMENTS

Authors acknowledge Ngāi Tūāhuriri as the University of Canterbury's mana whenua partner, and the Potano (Timucua) and Seminole Tribes, on whose lands the University of Florida is located. Thanks are also due to the three reviewers and the handling editor for their constructive comments that significantly improved earlier versions of this manuscript.

### Funding

Esteban N. Rodofili is funded jointly by the School of Natural Resources and Environment and the School of Forest, Fisheries, and Geomatics Sciences (College of Agricultural and Life Sciences) of the University of Florida. Funds allocated to Vincent Lecours by the University of Florida Senior Vice President for Agriculture and Natural Resources were also used to support this work. Support for Michelle LaRue was provided by the School

of Earth and Environment, University of Canterbury. There was no additional external funding received for this study. The funders had no role in study design, data collection and analysis, decision to publish, or preparation of the manuscript.

### Grant Disclosures
The following grant information was disclosed by the authors:
School of Natural Resources and Environment.
School of Forest, Fisheries, and Geomatics Sciences (College of Agricultural and Life Sciences) of the University of Florida.

### Competing Interests
The authors declare there are no competing interests.

### Author Contributions
- Esteban N. Rodofili conceived and designed the experiments, performed the experiments, analyzed the data, prepared figures and/or tables, authored or reviewed drafts of the article, and approved the final draft.
- Vincent Lecours performed the experiments, analyzed the data, authored or reviewed drafts of the article, and approved the final draft.
- Michelle LaRue performed the experiments, analyzed the data, authored or reviewed drafts of the article, and approved the final draft.

### Data Availability
 This work is a literature review and is not data-driven.

### Supplemental Information
Supplemental information for this article can be found online at http://dx.doi.org/10.7717/peerj.13540#supplemental-information.

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
