# Peer review of "Remote sensing techniques for automated marine mammals detection: a review of methods and current challenges"

_PeerJ, doi:10.7717/peerj.13540_

## Round 0.1 · original submission · Major Revisions

I consider your work to be an interesting review of different methods for remote sensing of marine mammals.

The three reviewers suggest changes that should be taken into account, mainly reviewer two, mentioning the need to include key papers for the review work. I hope that the changes are incorporated and the paper resubmitted for publication.

·

Basic reporting

The analysis provides an updated review of the efficacy of different remote sensing methods for detecting marine mammals.
The review and its introduction are deep and limited to the objectives, and they are sufficiently innovative.
The review's focus is of broad and cross-disciplinary interest, so is within the journal's scope.

Experimental design

• The review has been performed through automated bibliographic searches and can be replicated. Sources are appropriately cited.

Validity of the findings

Sources are appropriately cited.
The description of satellite data sources is not sufficiently developed. Because this is a review, the reader must access more information to compare and decide on the available methods and techniques for a more helpful review.
Satellite and UAS information could be improved, i.e., by adding a second table with each image's spatial and spectral resolution characteristics or identifying which satellite platform or UAS the data is being taken from (e.g., Worldview 2 is named in one case).
If the platforms are not identified in the papers analyzed, they should be made explicit in the review.
On the other hand, since Marine mammals species are very different in size, it would be necessary to identify the spatial resolution of each study.

Additional comments

About the title:
Replacing the title with one similar to: 
"Remote sensing techniques for marine mammals detection: a review of methods and current challenges of automated approaches" could better reflect the content of the work.
The topics analyzed are correctly exposed, but this review would be desirable to contain more information from each reviewed paper.
Particular comments:
In lines 442-449, the authors discuss the importance of analyzing the spatial resolution and the spectral bands of sensed imagery for better detection, but the manuscript does not provide data on this. This aspect could be improved i.e., by adding a second table.

Lines 472-473: Please provide references on studies of passive acoustic monitoring for submerged animals and satellite tracking for detailed movement patterns.

Although the original works do not explicit this, Google Earth comes from "processed" satellite images classified as such. Furthermore, there is a reference to which image is used in each Google earth view.

·

Basic reporting

The introduction and background supporting the rationale for the review is weak. While it is stated who this review is targeted towards, there is little evidence provided as to why the paper is needed and how it differs from similar previous reviews in the field, i.e.;
• Hollings et al., 2018
• Wang et al., 2019 – Surveying Wild Animals from Satellites, Manned Aircraft and Unmanned Aerial Systems (UASs): A Review
• Corcoran et al., 2021 – Automated detection of wildlife using drones: Synthesis, opportunities and constraints
The background needs more detail on the use of automated methods for detecting animals in satellite and UAS imagery (including terrestrial animals), the types of methods, and their increasing use along with their potential advantages. This would provide the rationale for the review (for example, relatively little use of automated methods in marine mammals) and the rest of the background can then be more briefly summarised. The readability and flow of the paper would be greatly improved if the background context emphasized that manual count of animals in satellite and UAS imagery is time and labor-intensive, but that automated methods of image analysis offer promising solutions.

Experimental design

The literature search does not appear to be rigorous. There was an important seminal paper missing:
• Maire et al. (2015) – Automating Marine Mammal Detection in Aerial Images Captured During Wildlife Surveys: A Deep Learning Approach
As well as two other more recent papers:
• Gray et al. (2018) - A convolutional neural network for detecting sea turtles in drone imagery
• Woolcock et al. (2021) - Effectiveness of using drones and convolutional neural networks to monitor aquatic megafauna
Ideally, the literature search should be conducted with a database such as Web of Science or Scopus, in addition to Google Scholar. Inclusion of other keywords such as “drone”, “automat*”, and “detect*” etc may increase the number of related papers.
Line 156 – “…works that compared detections with imagery of different resolutions.” – it is not clear what this means and what studies this criterion excludes.
The reasons for the inclusion of studies of marine mammal detection using satellite or UAS imagery that did not include automated methods for detection is not stated and is unclear. Since the paper is primarily focused on the use of automated methods, it seems appropriate that the reviewed papers and Table 1 only include those that apply automated methods.

Validity of the findings

The broad recommendation of the potential use of thermal imagery for marine mammal detection should be cautioned. The thermal imaging study referenced (Seymour et al., 2017), although detects pinnipeds, is conducted primarily on a terrestrial background. The use of thermal imagery in in-water marine environments is not discussed in the paper.
Gonçalves, Spitzbart & Lynch (2020) incorrectly define the F1-score, I would not use this paper to support the argument that metrics are lacking.
In spite of my agreement that there is a lack of consensus regarding reporting metrics, I remain skeptical about the novelty and utility of the proposed evaluation metrics and what they will add to the field of automated detection. For example, Equation 1 is commonly known as miss rate or false negative rate (FNR), and it is no more informative than recall, since it can be calculated directly from it. In the same way, Equation 2 doesn't seem any more informative than precision since it can indirectly tell us the number of false positives or "overcounts" per correct detection (true positive). I am open to the idea, but there needs to be greater justification for why practitioners should use the proposed evaluation metrics over more commonly used ones such as recall, precision, and average precision - as evidenced across computer vision and remote sensing object detection studies.
Line 291 - “Table 1 suggests that this may not be true in the field of marine mammal detection.” – what exactly in Table 1 suggests?
Re: Table 1 - Dujon et al. (2021) reports confusion matrix in their supplementary material. Equations 1, 2 and 3 can be calculated.
Line 378 – “Overall, we concur with Hollings et al. (2018) argument that automation is a viable alternative to reduce the time and labor necessary to analyze remote sensing imagery,…”. The paper does not provide hard evidence of a reduction in labor or time in the analysed studies. Although, line 391 does reference Thums et al. (2018), which suggests time savings with automated methods - this could be moved further up and expanded on. A further comparison of manual analysis vs automated analysis in the context of time savings across the reviewed studies would be needed to support the findings of Hollings et al. (2018).

Additional comments

The paper seeks to review the literature for automated methods applied to detect marine mammals using satellite and UAS imagery. To my knowledge, the topic of this paper is novel, and it would be the first paper to synthesize and compare automated methods applied to marine mammals.
When I first read the abstract, I thought the description and summary of the paper sounded intriguing and well-organized, but as I read the full report, this feeling faded somewhat. I would suggest the paper has potential, however, revision through addressing at least the above comments should be considered before acceptance.
It is important, I think, to emphasize that manual counting of animals in satellite and UAS imagery is time-intensive and labor-intensive and that automated methods offer promising solutions. It is one thing to have these automated methods, but it is another to convince researchers and practitioners of their utility.
It is apparent there is limited research on automated methods for satellite and UAS imagery for marine mammal detection, and some further comments on possible reasons for this would be helpful to the reader.

Reviewer 3 ·

Basic reporting

This is an interesting and well approached review of the use of UAS and satellite imagery to detect and count marine mammals. It adds results of novel tools to previous reviews on these subjects.
I have only some minor comments:
Lines 66-67 should also mention the technique of counting animals while surrounding rookeries on boats, which is also very common.
Maybe the first study that counted pinnipeds from UAS imagery and compared them to those made by eye from boats is Adame et al. (2017). I think it deserves a mention in the Introduction.
Line 162. I think the authors should use an alternative strategy to communicate the patterns they claim from Table 1. I encourage them to run a simple metanalysis of the bibliographical results so they can draw such conclusions.
Lines 212-212: I think the imperfect detection of cetaceans deserves more discussion. This is important, because no matter how good a technique is detecting cetaceans at surface from remotely sensed imagery, some probability of detection should be possible to estimate to truly replace ship- and airplane-based surveys for cetacean density and abundance estimations. Is there any idea in the literature to deal with this? Is there any potential development in that field?
Line 225: I think the indices you mention need to be as main equations within the text instead of just footnotes.
Line 302: Please add a number next to the equations within the text.
Lines 302-306: I do not see the contribution of the authors’ proposed index. It is just the complementary proportion of the index explained before. Is it just semantics o is there a real reason for that?
Since the authors are proposing a new index of the proportion of animals missed by the counting technique and mention others previously proposed, I think they should present the application of those indices to the results of the studies they compiled in plot, so the reader can compare their performance.

Reference:
Adame K, Pardo MA, Salvadeo C, Beier E, Elorriaga-Verplancken FR. 2017. Detectability and categorization of California sea lions using an unmanned aerial vehicle. Marine Mammal Science 33:913–925. DOI: 10.1111/mms.12403.

Experimental design

'no comment'

Validity of the findings

'no comment'

Additional comments

'no comment'

---

## Round 0.2 · accepted · Accept

Thank you very much for submitting your paper again. I believe you have made the changes suggested by the reviewers, including bibliography, improving the tables, rewriting the sections suggested by the reviewers, and reordering and clarifying the discussion. I think the paper is ready to be published.